# Electrochemical Sensing of Pb^2+^ and Cd^2+^ Ions with the Use of Electrode Modified with Carbon-Covered Halloysite and Carbon Nanotubes

**DOI:** 10.3390/molecules27144608

**Published:** 2022-07-19

**Authors:** Paweł Knihnicki, Aleksandra Skrzypek, Małgorzata Jakubowska, Radosław Porada, Anna Rokicińska, Piotr Kuśtrowski, Paweł Kościelniak, Jolanta Kochana

**Affiliations:** 1Department of Analytical Chemistry, Faculty of Chemistry, Jagiellonian University, Gronostajowa 2, 30-387 Krakow, Poland; aleksandra.skrzypek@student.uj.edu.pl (A.S.); pawel.koscielniak@uj.edu.pl (P.K.); jolanta.kochana@uj.edu.pl (J.K.); 2Faculty of Materials and Ceramics, AGH University of Science and Technology, A. Mickiewicza 30, 30-059 Krakow, Poland; jakubows@agh.edu.pl (M.J.); rporada@agh.edu.pl (R.P.); 3Department of Chemical Technology, Faculty of Chemistry, Jagiellonian University, Gronostajowa 2, 30-387 Krakow, Poland; anna.rokicinska@uj.edu.pl (A.R.); piotr.kustrowski@uj.edu.pl (P.K.)

**Keywords:** halloysite, sensor, cadmium, lead

## Abstract

A novel voltammetric method for the sensitive and selective determination of cadmium and lead ions using screen-printed carbon electrodes (SPCEs) modified with carbon-deposited natural halloysite (C_Hal) and multi-walled carbon nanotubes (MWCNTs) was developed. The electrochemical properties of the proposed sensor were investigated by electrochemical impedance spectroscopy (EIS) and cyclic voltammetry (CV), while the morphology and structure were established by scanning electron microscopy (SEM) and X-ray powder diffraction (XRD). A two-factorial central composite design (CCD) was employed to select the composition of the nanocomposite modifying the electrode surface. The optimal measuring parameters of differential pulse anodic stripping voltammetry (DPASV) used for quantitative analysis were established with the Nelder–Mead simplex method. In the analytical investigation of Cd(II) and Pb(II) ions by DPASV, the MWCNTs/C_Hal/Nafion/SPCE exhibited a linear response in the concentration range of 0.1–10.0 µmol L^−1^ (for both ions) with a detection limit of 0.0051 and 0.0106 µmol L^−1^ for Pb(II) and Cd(II), respectively. The proposed sensor was successfully applied for the determination of metal ions in different natural water and honey samples with recovery values of 96.4–101.6%.

## 1. Introduction

The determination of heavy metal content in the environment is of great importance since ions of such metals (e.g., mercury, cadmium, lead, chromium and arsenic) have a detrimental effect on human health [1]. The accumulation of lead cations in the human body can influence children neurobehavioral development, contributing to an increase in blood pressure, anemia and kidney injury. Cadmium cations are known as strong carcinogenic agents responsible for lung cancer, osteomalacia and proteinuria [2].

Heavy metals are released into the environment from various natural and anthropogenic sources. The latter include the developed mining industry, production of batteries, pigments, alloys, electroplating and coating. Due to their long-term chemical stability, non-degradable properties, rapid movement in the soil–plant–human trophic chain, as well as the tendency to accumulate in the environment and living organisms, they constitute a challenge for global sustainability [2]. The determination of heavy metals remains a difficult task, since they are present at low concentration (<2 ppb) in samples of complicated matrices such as animal tissues, food products, waste water, soil and plants [1]. For this reason, highly sensitive analytical methods for the determination of toxic metal ions, including Pb^2+^ and Cd^2+^ ions, are constantly required. Atomic absorption spectrometry (AAS) [3,4] and inductively coupled plasma mass spectrometry (ICP-MS) [5,6] have been reported for this purpose; however, electrochemical methods are most commonly used. Special attention has been paid to the latter techniques due to their simplicity, sensitivity, no need for complicated sample preparation, as well as low instrumentation and analysis costs. Moreover, they can constitute a core of portable analytical systems for point-of-care or on-site analysis in analytical chemistry [7]. Because of their simplicity, resulting from the possibility of integrating three electrodes on a small surface, low power requirement and low cost of production, screen-printed electrodes (SPE) are the best choice. Moreover, the necessity to quantify trace concentrations of heavy metals and simultaneously a small surface of the working electrode requires the electrodes’s surface to be modified in order to improve the analytical parameters of the sensor [1]. The simultaneous electrochemical determination of heavy metals using a sensor based on a nanocomposite consisting of fluorinated graphene and gold nanocage was reported by Zhao [8]. A disposable electrode based on graphene paper–Nafion–Bi nanostructures for ultra-trace analysis of Pb^2+^ and Cd^2+^ has been proposed [9]. Hassan et al. developed a bimetallic Hg-Bi composite supported on a poly(1,2-diaminoanthraquinone)/glassy carbon-modified electrode for the single and simultaneous voltammetric sensing of Pb^2+^, Cd^2^+ and Zn^2+^ [10]. The orthorhombic aluminum ferrite phase modified glassy carbon electrode allowed for the simultaneous sensing of copper, lead, cadmium and mercury traces in human blood serum [11].

In recent years, various nanocomposites have been extensively studied for the adsorption, detection and removal of heavy metal ions present in aqueous solutions or soils, especially employing electrochemical methods. Among them, clay minerals have gained a special interest due to their unique features such as high specific surface area, natural availability at low price and low toxicity. Halloysite nanotubes are particular clay minerals with a predominantly hollow tubular structure, which shows a peculiar chemical surface composition allowing for the selective functionalization at the inner and outer surface [2]. For instance, silver nanoparticles incorporated in dicarboxylic/TEPA modified halloysite nanotubes were reported for the degradation of organic contaminants [12]. Modified halloysite nanotubes and graphene oxide nanosheets were used for the fabrication of polycarbonate ultrafiltration mixed matrix membranes for olive oil/water emulsion separation [13]. For the electrochemical biosensing of hydrogen peroxide, a humic acid/halloysite nanotube/flavin adenine dinucleotide nanocomposite was proposed [14].

The aim of this work was to develop a sensitive electrochemical method for the simultaneous determination of Pb^2+^ and Cd^2+^ cations using the screen-printed carbon electrode (SPCE) modified with C-deposited halloysite (C_Hal), carbon nanotubes and Nafion. Due to its structure, the halloysite was used to increase the electrode surface and to improve contact between the electrode surface and the analyte. In order to provide electrical conductivity, a natural halloysite was covered with a carbon layer, which was formed from the decomposition of methane catalyzed by Fe species naturally occurring in the clay. The role of the carbon nanotubes (MWCNTs) was to enhance the conductivity of the electrode composite, while Nafion acted as a binder. A two-factorial central composite design (CCD) was employed to select the optimal composition of the nanocomposite modifying the electrode surface. Electrochemical impedance spectroscopy (EIS) and cyclic voltammetry (CV) were employed for the electrochemical characterization of the proposed MWCNTs/C_Hal/Nafion/SPCE. The structural features of the composite were investigated using X-ray powder diffraction (XRD), and the morphology of the electrode composite was studied by scanning electron microscopy (SEM). The optimal measuring parameters of differential pulse anodic stripping voltammetry (DPASV), which was used for quantitative analysis, were established with the Nelder–Mead simplex method. The analytical characteristics of the sensor were carried out, and the verification of the developed sensor’s performance was successfully performed by determining Cd^2+^ and Pb^2+^ ions in tap, mineral and well water samples as well as in a honey sample. 

## 2. Results and Discussion

### 2.1. Choice of Sensor Modification Composite. Preliminary Study

In the first step of selecting the composition of the electrode modifying mixture, cyclic voltammetry measurements were carried out for the tested sensor. The following electrodes were taken into account: SPCE, C_Hal/Nafion/SPCE, MWCNTs/C_Hal/Nafion/SPCE and CMK-3/C_Hal/Nafion/SPCE. For the preliminary measurements, the concentration of C_halloysite, MWCNTs, and CMK-3 was constant and equal to 1% (*w*/*w*). In this part of the experiments, three concentrations of Nafion were tested: 0.5%, 1.0% and 1.5% (*w*/*w*). At the lowest concentrations of Nafion, the composite was too weakly bonded to the electrode surface, and it was removed after immersing that electrode in the measuring solution. For the Nafion concentration of 1.5%, no increase in the signal was observed, so for further research, the mixture containing 1.0% (*w*/*w*) Nafion was selected. For the tested electrodes, CV measurements in the presence of Fe^2+^/Fe^3+^ ions were carried out using electrode *polarization* rates from 2 to 300 mV s^−1^ (see Appendix A). For all the studied sensors, the relationship between the intensity of the oxidation current and the root of the scan rate was linear, as evidenced by the high value of the coefficient of determination R^2^. The linearity of the dependence *I = f(ν^½^)* (see Appendix A) indicated that the processes of Fe^2+^/Fe^3+^ redox probe on the tested electrodes occurred only through diffusion, and the diffusion current was measured.

Based on the recorded curves, the electrochemical active surface areas were estimated using the Randles–Ševčik equation:$$I_{p}=kz^{2/3}D^{1/2}A_{el}\upsilon^{1/2}c$$
where *k*—constant: 269,000, *z*—the number of electrons involved in the electrode reaction, the value of the diffusion coefficient used for the calculations was equal to 7.6 × 10^−6^ (cm^2^ s^−1^) [15], $A_{el}$—electrode area (cm^2^), υ—scan rate (V s^−1^), c—concentration of the depolarizer in the depth of the solution (mol cm^−3^).

The unmodified screen-printed carbon electrode exhibited the smallest electrochemical surface area, 3.34 mm^2^. Modification of the electrode with C_halloysite suspended in Nafion increased the active surface to 4.11 mm^2^. An additional enrichment of the suspension with the carbon materials, MWCNTs or CMK-3, resulted further in its rise, to 7.25 and 5.65 mm^2^, respectively. The geometric electrode area (for unmodified electrode 7.07 mm^2^) assumes that the electrode surface is flat and can be calculated based on the size and shape of the electrode. The real electrode surface takes into account the roughness and porosity of the interface, and the differential capacity of the electrical double layer is usually proportional to it. The electroactive area is the area that is active for the charge transfer across the electrode–electrolyte interface and is estimated in electrochemical experiments for model redox systems [16]. Thus, the electroactive surface is always equal or lower than the real electrode surface, which in turn is usually higher than the geometric area. However, in the case of screen-printed electrodes, the electroactive surface area lower than the geometric area has been reported by other researchers [17], which most probably results from the method of their production (mixing of a conducting carbon material with nonconducting polymeric matrix).

Therefore, the sensor modified with the composite containing 1% C_Hal and 1% MWCNTs in 1% Nafion solution (all % *w*/*w*) (MWCNTs/C_Hal/Nafion/SPCE) was chosen for further research.

### 2.2. Application of CCD Method

A two-factorial central composite design (CCD) [18,19,20] was employed to select the composition of the electrode surface modified with C-deposited halloysite and multi-walled carbon nanotubes. In preliminary experiments, the contents of MWCNTs (factor *x*_1_) and C_Hal (factor *x*_2_) were found to influence the operation of the sensor. Considering previous observations, it was assumed that each of the two variables would be changed in the range of values from 0.5% to 1.7% (*w*/*w*). The optimization criterion was the sensitivity of the voltammetric method for the determination of cadmium (y_Cd_) and lead (y_Pb_), as these two analytes were the subject of interest in this work. The research was conducted in such a way as to obtain maximum sensitivity. A series of experiments, designed according to the CCD, were performed independently optimizing the addition of nanocomposites in the construction of electrodes for cadmium and lead determination. Table 1 presents the values of the actual and coded variables used in the optimization experiments. This is a list of 13 experiments consisting of 4 points from the full factorial design, 4 star points, and 5 repetitions at the central point.

Using the list given in Table 1, 13 sensors were constructed, and the operation of each of them was tested in such a way that the calibration relationship for cadmium and lead was recorded in the concentration range of 0–10 μmol·L^−1^. The area under the peak was taken as the analytical signal. In each case, a satisfactory correlation between the analytical signal and concentration was obtained, meeting the typical acceptance criteria in analytical chemistry (r > 0.995). The obtained sensitivity values, separately for cadmium and lead determination, were the basis for the definition of a quadratic equation (in accordance with the CCD assumptions), approximating the dependence of the response (y_Cd_, y_Pb_) from the variables x_1_ and x_2_. The general form of this second-order equation is given by the following formula:y = b_0_ + b_1×1_ + b_2_x_2_ + b_3_x_1_^2^ + b_4_x_1_x_2_ + b_5_x_2_^2^

This relationship is typically interpreted as the response surface equation [21,22]. The calculated values of the b_0_ to b_5_ coefficients for these two models, along with the assessment of their significance, are presented in Appendix A.

During the construction of the model, the aim of which was to optimize the electrode modification composite to obtain the maximum sensitivity of determination, r^2^ was estimated, which is the assessment of data compliance with the model. In the case of cadmium, r^2^ was 0.921, while for lead, an r^2^ value of 0.902 was obtained.

The graph of the response surface for the sensor optimization related to cadmium and lead determination is presented in Figure 1. It is important that in the considered range of components content, the response surface reaches the maximum (Figure 1a,c). The position of the maximum indicates the optimal sensor composition. For the determination of cadmium, the function reaches its maximum at the point %MWCNTs = 1.03, %C_Hal = 1.18, whereas for lead—at the point %MWCNTs = 0.82, %C_Hal = 1.00. Figure 1b,d show a high agreement of the fit of response surface to the data, as the sensitivity values calculated from the response surface equation (predicted) are consistent with the experimental values. The 13 pairs of bars in the graph correspond to the 13 sensors made.

Because the aim of the work was to build one sensor for the optimal determination of cadmium and lead, one figure with the response surfaces representing both models was created. The values lying at the intersection of the two bold ellipses (Figure 2) indicate an optimal composition of the sensor surface. On this basis, the optimal contents of the tested nanocomponents in electrode composite were selected and used in the further experiments: 1.05% (*w*/*w*) C_Hal and 0.95% (*w*/*w*) MWCNTs suspended in a 1% (*w*/*w*) Nafion solution.

### 2.3. Composite Characterization

#### 2.3.1. Structural and Textural Characterization of Halloysite

The phase composition of halloysite-based materials (before and after the C layer deposition) was examined with X-ray powder diffraction (Figure 3A). The diffraction pattern of the raw clay mineral shows reflections characteristic of the halloysite structure. It is commonly known that natural halloysite contains a variable amount of iron depending on its origin. In the case of the studied clay mineral, it was approx. 14 wt % of Fe. Due to the presence of iron in the natural halloysite, the diffraction peaks at 33.3° (104), 35.7° (110), 49.5° (024) and 54.2° (116) [PDF 00-001-1053] related to the Fe_2_O_3_ phase are additionally detected. The pre-reduction step followed by the methane decomposition resulted in the appearance of metallic Fe and Fe_3_C phases instead of Fe_2_O_3_. In the C_Hal sample, three additional diffraction peaks at 26.7°, 43.1°, and 43.8°, corresponding to the graphite structure (for example, with the inter-shell spacing of the concentric cylinders of graphitic carbon [23]), are observed.

The low-temperature nitrogen adsorption–desorption isotherm of C_Hal is shown in Figure 3B. A certain increase in adsorption effect within the whole p/p_0_ range as well as a hysteresis loop (type H3 according to the IUPAC classification) [24] is identified. The C_Hal exhibits a BET surface area of 47 m^2^ g^−1^ and a total pore volume of 0.15 cm^3^ g^−1^. It can therefore be concluded that the porosity of the studied material comes mainly from the halloysite support (practically without microporosity typical for chosen carbons), which generates the expected meso- and macroporous system. On the other hand, the deposition of graphite structures by the catalytic decomposition of methane did not lead to a complete degradation of porosity.

#### 2.3.2. Morphological Characterization of Electrode Nanocomposite

SEM images (Figure 4) show differences in the morphology of the electrode’s surface for unmodified SPCE (Figure 4A), SPCE modified with C_Hal and Nafion (Figure 4B), and SPCE modified with C_Hal, Nafion and MWCNTs (Figure 4C). The unmodified electrode is characterized by the highest composition homogeneity, but this surface is also not free of small inclusions. The introduction of C_halloysite visibly increased the surface area in relation to the unmodified electrode. The most porous structure was obtained after the modification of the electrode surface with the mixture containing carbon nanotubes and C_halloysite. The increase in porosity due to the introduction of aluminosilicate material and carbon nanotubes resulted in an increase in the surface area capable of reducing the determined cations.

#### 2.3.3. Electrochemical Characterization of MWCNTs/C_Hal/Nafion/SPCE

To examine the electrochemical properties of the sensor modified with the selected composite (unmodified SPCE, MWCNTs/Nafion/SPCE and MWCNTs/C_Hal/Nafion/SPCE), electrochemical impedance spectroscopy was employed. In order to determine the quantitative parameters characterizing the processes of charge transfer, mass transport and charging of the electrical double layer, an equivalent circuit was used. The recorded impedance spectra for all tested electrodes, in the form of a Nyquist plot, are shown in Figure 5.

The dashed line marks the characteristic semicircle representing the charge transfer resistance and charging of the electric double layer, which was obtained as a result of fitting the model spectrum to the high-frequency fragment of the spectrum. During the fitting of an model impedance spectrum to the measuring values, the Randles equivalent circuit has been used. It consists of the solution resistance (Rs), constant phase element (CPE), charge transfer resistance (Rct), and Warburg impedance (ZW). CPE models the electrical double layer at the electrode–electrolyte interface, whereas Rct is the measure of the ease of electron transfer across this phase boundary. ZW represents the impedance of the depolarizer diffusion toward the electrode surface, and it is inversely proportional to the electrode surface. The fitting has been performed taking into account the whole impedance spectra, which was recorded in the frequency range of 100 kHz to 25 mHz. For the tested electrodes, that is SPCE, MWCNTs/Nafion/SPCE, and MWCNTs/C_Hal/Nafion/SPCE, the spectra are composed of a semicircle and a straight line in the high and low frequency range, respectively. However, due to the high capacity of the electrical double layer, the semicircle is not easily distinguishable in the case of the modified electrodes. Therefore, we have added the simulated spectra (dashed lines in Figure 5), for which the parameters of each circuit element were taken from the performed simulation, with the exception of the ZW, which was set to 0. This measure aimed at highlighting the influence of the chargé-transfer kinetics.

The values of differential capacity of the double layer (*C_dl_*), charge transfer resistance (*R_ct_*), and Warburg impedance module (σ) obtained as a result of fitting the model spectra as well as the effective electrode surface (*A_eff_*) and heterogeneous rate constant of the electrode reaction (*k_s_*) determined on their basis are presented in Table 2.

The MWCNTs/C_Hal/Nafion/SPCE sensor was characterized by the lowest charge transfer resistance among all the tested electrodes. For this electrode, the determined rate constant was two orders of magnitude greater than for the other sensors, indicating slight kinetic constraints. Furthermore, the value of the Warburg impedance module *σ* is approx. 70 times greater than *R_ct_*, which means that the process taking place on this sensor was reversible (the Nyquist diagram consists mainly of a straight line responsible for diffusion), in which depolarizer diffusion was the slowest step and determined the observed reaction rate. In the case of the remaining electrodes, the Nyquist diagram shows both a semicircle representing the process of charge transfer and charging of the electrical double layer and a straight line related to the diffusion of the depolarizer; additionally, the corresponding values of *R_ct_* and *σ* are comparable, which indicates the quasi-reversible nature of the electrode process.

The differential capacity *C_dl_* of the electrical double layer has been calculated from the following formula [25,26]:$$C_{dl}=\frac{1}{A_{geom}}\cdot \sqrt[{q}]{Q\cdot R_{ct}^{1-q}}$$
where *A_geom_*—geometric area of the electrode, *Q*, *q*—modulus and exponent of the constant phase element (a generalized “version” of a capacitor), and *R_ct_*—charge transfer resistance. The value of *C_dl_* depends on the real electrode surface and the adsorption properties at the phase interface. Thus, changes in *C_dl_* cannot be unequivocally ascribed to the surface development as a result of the used modifier(s). The structure of the MWCNTs is well suited for the incorporation of miscellaneous ions and molecules [27]; therefore, the use of carbon nanomaterials results in the increase in *C_dl_*, which translates into a higher value of the background current recorded in the voltammetric experiments. The utilization of the carbon-deposited halloysite of improved electrical conductivity contributes to the increase in electrode surface and more homogenous distribution of the MWCNTs on the electrode surface, avoiding the formation of MWCNTs conglomerates. This explains why the electroactive surface of the MWCNTs/C_Hal/Nafion/SPCE is higher than for MWCNTs/Nafion/SPCE. A similar decrease in the *C_dl_* of the electrode modified with nanomaterials (MWCNTs, AuNPs) after the introduction of micrometric materials (metal-organic frameworks or zeolites) to the modifying layer has been also noted by other researchers [28].

### 2.4. Optimization of Experimental Parameters of Cd^2+^ and Pb^2+^ Cations Determination

#### 2.4.1. Selection of Supporting Electrolyte

In the next step, several supporting electrolytes were tested for the simultaneous determination of Pb^2+^ and Cd^2+^ ions: acetate buffer (concentration: 0.1 mmol L^−1^, pH 4.0, 4.5 and 5.0; concentration 1 mmol L^−1^, pH 4.0, 4.5 and 5.0), 0.1 mmol L^−1^ phosphate buffer (pH 5.0, 6.0 and 7.0), 0.1 mmol L^−1^ ammonium buffer (pH 10.0) and 0.1 mmol L^−1^ sodium nitrate. All experiments were carried out using cyclic voltammetry at the scan rate 100 mV s^−1^ within the concentration range of both analytes 1–10 µmol L^−1^. The obtained results are summarized in Appendix A. The use of a 0.1 mol L^−1^ phosphate buffer (pH 5.0/6.0/7.0) solution as the supporting electrolyte allowed the observation of oxidation peaks derived only from cadmium, and in the case of using the 0.1 mol L^−1^ NaNO_3_ solution, peaks derived only from lead were revealed. For the 0.1 mol L^−1^ ammonium buffer solution (pH 10.0), no peaks were observed. The optimal supporting electrolyte, which made it possible to obtain well-shaped peaks from both ions, was 0.1 mol L^−1^ acetate buffer at pH 4.0. Increasing the acetate buffer concentration (pH 4.0) to 1 mol L^−1^ resulted in registering better shaped peaks with higher intensity; therefore, this buffer was selected as the supporting electrolyte for further study. Potassium chloride was used as one of the components of the supporting electrolyte in order to ensure the appropriate concentration of chloride ions (required in connection with the use of a screen-printed reference electrode). The concentration of chloride ions was tested in the range of 10–100 mmol L^−1^, and a concentration of 50 mmol L^−1^ was chosen as optimal.

#### 2.4.2. Optimization of DPASV Parameters Using the Nelder–Mead Simplex Method

As the analytical tool more sensitive than cyclic voltammetry, the differential pulse technique was employed. The optimization of DPASV parameters for quantitative measurements using MWCNTs/C_Hal/Nafion/SPCE was performed with the Nelder and Mead simplex method [29,30]. The following parameters were optimized: step potential—E_s_, pulse potential—E_p_, pulse time—t_p_ and polarization rate (scan rate) of the working electrode—υ. It was assumed that the pulse potential should not exceed 250 mV, the step potential should not exceed 20 mV, and the pulse time should not exceed 30 ms. Calibration curves based on the DPASV measurements carried out in solutions containing both Cd^2+^ and Pb^2+^ ions at micromolar concentrations were prepared, from which the values of sensitivity were compared as the response function (optimization parameter). Determination factor R^2^ was selected as an auxiliary criterion. The optimization process of DPASV parameters was carried out for 21 optimization points, obtaining 11 simplexes. Appendix A summarizes the measurement parameters obtained during the optimization, and Figure 6 shows the sensitivity changes for each optimization point created in the simplex process. Taking into account the obtained results, conditions corresponding to point 16 were selected as the optimal ones. The measurements carried out for the consecutive points either did not bring an increase in the measured signal or were impossible to realize owing to assumed parameter values.

Thus, the optimization process allowed selecting the best values of parameters, which were E_p_ = 150 mV, E_s_ = 15 mV, t_p_ = 25 ms and υ = 15 mV s^−1^. Consequently, further studies were performed by the differential pulse technique using these experimental parameter values.

#### 2.4.3. Study of Deposition Potential and Time (E_dep_, t_dep_)

The deposition of analyte molecules on the working electrode surface is a very important step to achieve the best response in the voltammetric analysis. Therefore, the effect of a deposition potential on the stripping peak currents was studied in the potential range of −0.5 to −1.5 V with the established time of analytes’ accumulation on the electrode equal to 60 s. As shown in Appendix A, the slope of the calibration curves for Pb^2+^ and Cd^2+^ slightly increased starting from the applied potential of −500 mV and reached a maximum at potential of −1.2 V. On the other hand, when the deposition potential was more negative than −1.2 V, a decrease in sensitivity values was observed. This phenomenon can be explained by the evolution of hydrogen at more negative potentials [31], so the deposition of metal ions on the electrode surface could be difficult due to the presence of visible hydrogen bubbles on the electrode surface. Hence, −1.2 V was selected as the optimal potential. In the next step, the deposition time was tested in the range from 30 to 240 s with a step of 30 s. The slopes of the calibration curves (obtained for an anodic peak current for micromolar concentrations of both ions) increased with a change of deposition time up to 90 s, above which they remained nearly constant and took lower values (see Appendix A). During the research related to the examination of the optimal deposition potential and time, the influence of stirring on the obtained results was also investigated. The effect of mixing for both types of ion was compared as the slope values of the calibration curves. Comparing the obtained sensitivity values, it was concluded that stirring the solution during the deposition of metal ions favorably affected the sensitivity observed for Pb^2+^ and Cd^2+^ ions. Therefore, a deposition time (t_dep_) of 90 s at potential (E_dep_) −1200 mV and stirring during accumulation were selected for further experiments as optimal conditions.

#### 2.4.4. Analytical Characteristics of MWCNTs/C_Hal/Nafion/SPE toward Cd^2+^ and Pb^2+^ Cations

The developed MWCNTs/C_Hal/Nafion/SPCE sensor was used for the simultaneous determination of cadmium and lead ions. Under optimal conditions, adsorptive stripping voltammograms were recorded for both analytes (Figure 7). The peak currents were proportional to the concentration of cadmium and lead in the 0.1–10 μmol L^−1^ concentration range, with LODs of 0.0106 and 0.0051 μmol L^−1^ for Cd^2+^ and Pb^2+^, respectively. The limit of detection (LOD) was calculated based on the formula: LOD = 3s/a, where: s—standard deviation of the mean for 100 nmol L^−1^ concentration of standard solution of each cation, a—the sensitivity of the calibration curves for cadmium or lead [32]. The average sensitivity (*n* = 3) was 20.70 µA L µmol^−1^ for Pb^2+^ and 6.27 µA L µmol^−1^ for Cd^2+^ with the repeatability of 3.3 and 2.8%, respectively (*n* = 3).

The fabrication reproducibility was tested for twelve MWCNTs/C_Hal/Nafion/SPCEs by comparing the oxidation peak currents of 0.1 µmol L^−1^ Cd^2+^ and Pb^2+^ cations. The relative standard deviation (RSD) was 5.7%, revealing that the proposed method of sensor preparation is characterized by good reproducibility. The long-term stability of the electrode functioning was also investigated by measuring the electrode response in solution containing 0.1 µmol L^−1^ Cd^2+^ and Pb^2+^ every 5 days. Between the measurements, the electrode was stored at room temperature in a closed box.

The current responses decreased to 93% for Pb^2+^ and 96% for Cd^2+^ after 10 days, while 90% of the original response was retained after 15 days for both cations. The electrode retained 87% of its original response even after 20 days.

The operational stability of the sensor was investigated for one electrode during 30 times repeated measurements for both cations in the solution containing Cd^2+^ and Pb^2+^ ions in a concentration of 0.1 µmol L^−1^. A recorded current decrease was observed for both cations, ca 5% and 3.5% for cadmium and lead, respectively, and the largest decline in drop of signals was observed for the first three to four measurements. On the basis of the conducted research, it can be concluded that the proposed electrode shows high stability in subsequent measurements, and therefore, it can be used many times in electrochemical analyses.

Table 3 compares the analytical response characteristics of our sensor with previously reported sensors developed for the determination of Pb^2+^ and Cd^2+^ cations.

In comparison to the previously reported electrochemical methods of Cd/Pb determination, the proposed MWCNTs/C_Hal/Nafion-modified screen-printed carbon electrode exhibits a wide linear range and satisfactory LOD value.

#### 2.4.5. Real Sample Analysis

To illustrate the suitability for real sample applications, MWCNTs/C_Hal/Nafion/SPCE was employed for the detection of Cd^2+^ and Pb^2+^ ions in laboratory tap water, well water, bottled drinking mineral waters and honey samples. The analyses were carried out with the use of the standard addition method. All samples were checked before addition of the analytes for the presence of Cd^2+^ and Pb^2+^ ions by the proposed method. Next, all the samples were spiked with standard Cd^2+^ and Pb^2+^ solutions to obtain an expected concentration. The analyses were performed for two concentrations levels (see Table 4). The quantification procedure was repeated three times with the use of one sensor for three separately prepared samples. Recovery, confidence intervals (with significance level α = 0.05) and relative standard deviation (RSD) for each concentration were calculated (see Table 4).

The obtained recoveries were in the range from 96.4% to 101.6%, indicating that the proposed procedure is accurate and selective enough for practical application. Lower recovery values were obtained for the honey samples (for both concentration levels), which is due to a more complex matrix compared to the others. The mean relative standard deviation for all samples was 4.8%, and it was not higher than 9%. The highest RSD values were noted for the honey samples; also, the highest relative errors were obtained for recovery values. The achieved results prove the wide applicability of the proposed MWCNTs/C_Hal/Nafion/SPCE sensor, even for samples with a complex matrix.

## 3. Materials and Methods

### 3.1. Chemical and Materials

All chemicals and reagents were of analytical grade and were used as received without further treatment. Metal standard solutions were prepared by dissolving appropriate amounts of salts, lead nitrate, Pb(NO_3_)_2_, and cadmium nitrate tetrahydrate, Cd(NO_3_)_2_·4H_2_O (Avantor, Gliwice, Poland), in plastic volumetric flasks. Acetate buffer was prepared by dissolving an appropriate amount of sodium acetate and acetic acid (both purchased from Avantor, Gliwice, Poland), with pH adjustment using hydrochloric acid or sodium hydroxide solution (both purchased from Avantor, Gliwice, Poland). Nafion stock solutions of 5% (*w*/*v*, Aldrich Chemicals, Darmstadt, Germany) in a mixture of low weight aliphatic alcohol and water were diluted as described in the tested procedure. K_4_[Fe(CN)_6_] and K_3_[Fe(CN)_6_] in KCl solutions (purchased from Avantor, Gliwice, Poland) were used as the model redox reagents in CV and EIS measurements. Multi-walled carbon nanotubes—MWCNTs (Aldrich Chemicals, Darmstadt, Germany) and a CMK-3 carbon replica (synthesized in Department of Chemical Technology, Faculty of Chemistry, Jagiellonian University) were tested as potential carbon materials for the accumulation and deposition of metal cations on an electrode surface. CMK-3 is a type of carbon material that belongs to Ordered Mesoporous Carbons (OMCs). CMK-3 was synthesized in 2000 using the SBA-15 silica of the P6 mm symmetry as the template and sucrose as the carbon precursor [42].

Deionized water from arium^®^ UltraPure water system (Sartorius) with a resistivity of 18.02 MΩ cm was used for all experiments. All plastic materials and quartz tubes were cleaned with the use of HNO_3_ (10%) and thoroughly rinsed with UltraPure water each time before use.

Natural halloysite (supplied by Kopalnia Haloizytu Dunino Sp. z o.o) was doped with carbon using the catalytic methane decomposition. Briefly, the raw clay mineral was calcined at 550 °C for 4 h (heating rate = 10 °C min^−1^) in air atmosphere. Then, the obtained material was introduced into a tubular oven and pre-reduced in a H_2_ flow at 550 °C for 1 h. The reactor was purged with a N_2_ flow for 15 min. Finally, the nitrogen flow was switched to methane (18 mL min^−1^), and the temperature was raised to 750 °C (heating rate = 10 °C min^−1^). After 6 h of treatment at the final temperature, the reactor was cooled to room temperature under N_2_ atmosphere. The resulting material (C_Hal) contained 41.8 wt % of carbon, as determined by thermogravimetric analysis.

### 3.2. Apparatus and Measurement Procedures

All voltammetric measurements were carried out using a multipurpose electrochemical workstation PalmSens2 (PalmSens, Houten, The Netherlands) connected to a three-electrode system (ItalSens). The SPCE three-electrode cell (ItalSens) consisted of a carbon auxiliary electrode and a silver reference electrode, which were both screen printed on a polyester material. The working electrode (with a geometric area of 7.07 mm^2^) was either an unmodified carbon electrode or a carbon electrode modified with Nafion, C-deposited halloysite, and additionally with CMK-3 or MWCNTs. The measurements were performed in a plastic cell covered with a Teflon cover with a hole adapted to the size of the screen-printed electrode. Preliminary experiments to estimate the electrode surface area were carried out in 10 mmol L^−1^ K_3_[Fe(CN)_6_] with 0.1 mol L^−1^ KCl solution at scan rates from 2 to 300 mV s^−1^.

To select the most suitable supporting electrolyte, measurements were carried out in 5 mL electrolyte solution with the addition of appropriate volumes of the analytes standard solution (Cd^2+^ and Pb^2+^ ions) to obtain their appropriate concentrations. For the supporting electrolyte solution and each of the respective analytes’ concentrations, voltammetric curves were recorded from which the current values of oxidation peaks were determined, and then, calibration curves were prepared, showing the dependence of the current on the concentration of analytes.

Differential pulse anodic stripping voltammetry (DPASV) with the deposition of Cd^2+^ and Pb^2+^ ions was employed to characterize the analytical properties of the developed sensor and to determine the analytes in the food samples. For this technique, measurement parameters, such as potential step, potential pulse, pulse time and scan rate, were optimized using the Nelder–Mead simplex method [25].

Impedance spectra were recorded using a µAUTOLAB III analyzer (EcoChemie, Utrecht, The Netherlands) with NOVA 2.0 software. The measurements were carried out using a three-electrode system consisting of a platinum wire (auxiliary electrode), a silver chloride electrode with a double jacket (reference electrode) and the tested electrode as the working electrode. The impedance spectra for 0.5 mmol L^−1^ solution [Fe(CN)_6_]^4−/3−^ dissolved in 1 mol L^−1^ KCl, acting as the supporting electrolyte, were recorded at the formal potential of the redox pair [Fe(CN)_6_]^4−/3−^ using sinusoidal signals with a frequency in the range 100 kHz ÷ 25 mHz and amplitude 10 mV. All these measurements were performed in a deoxygenated solution.

Powder X-ray diffraction (XRD) patterns were collected in a D2 Phaser (Bruker, Billerica, MA, USA) diffractometer with CuKα X-ray radiation (λ = 1.54184 nm) in the 2 θ range of 5–90° with step of 0.02°.

Textural properties of the C-deposited halloysite were determined by low-temperature adsorption of nitrogen at −196 °C using a Micromeritics ASAP 2020 sorptometer. Before measurements, samples were degassed at 250 °C for 5 h under vacuum conditions. Specific surface areas were calculated using the Brunauer–Emmett–Teller (BET) method. Total pore volumes were determined from amounts of nitrogen adsorbed at relative pressure p/p_0_ of ca. 0.97.

SEM measurements were performed using a VEGA3 (Tescan) scanning electron microscope with a LaB6 cathode equipped with SE (secondary electrons), BSE (back-scattered electrons). The SEM images were collected at an accelerating voltage of 30 kV.

### 3.3. Halloysite-Based Sensor Preparation

Before modification, SPCEs were gently purified with water to remove impurities. The working electrode was prepared by applying 5 μL of the appropriate modifying suspension onto its surface. The modifying mixture was composed of Nafion, C_Hal and carbon material: CMK-3 or MWCNTs. Each suspension was prepared and mixed immediately before use for about 2 min on a shaker. After deposing the modifying composite on the electrode surface, it was gently dried under compressed air for at least one minute. After preliminary experiments, which allowed to choose the components of the modification mixture, its optimal composition was selected using a two-factorial central composite design (CCD) [18].

### 3.4. Samples Preparation

Still mineral water samples were obtained from a local supermarket. The water samples differed in composition and overall mineralization (213.4 and 564.2 mg L^−1^). Tap water was collected in the laboratory; it was medium-hard water with pH of 6. The drinking water from the well (collected in the center of the Krakow) was characterized by a high iron content, as evidenced by the characteristic rusty coating at the collection site. The water from the well was described as hard (checked in our laboratory—21.2° N) by local authorities. Tap, mineral and well water samples were tested after mixing with acetate buffer to introduce the supporting electrolyte. Honey (from a local supermarket) was prepared according to the procedure described by Bougrini et al. [43]. Firstly, 1 g of honey was mixed with 1 mL of ethanol and maintained in an ultrasonic bath for half an hour. Then, to reduce the matrix effect, the sample was diluted 1/10 (*v*/*v*) with acetic buffer and filtered through a 0.8 mm filter. Before measurements, it was verified that none of the food samples contained cadmium or lead. Next, all the samples were spiked with standard Cd^2+^- and Pb^2+^-containing solutions to achieve expected concentrations of analytes.

## 4. Conclusions

In the present work, a novel screen-printed carbon electrode, based on carbon-deposited halloysite mixed with multi-walled carbon nanotubes, for the sensitive determination of Cd^2+^ and Pb^2+^ ions, was introduced. EIS and CV measurements confirmed the high surface area, enhanced electrical conductivity and improved analytical properties of the fabricated sensor in comparison to bare SPCE. The conducted research proved that the exceptional performance of the developed sensor is associated with the presence of natural clay, as the electrode surface modifier, in the electrochemical determination of metals. The use of natural material is part of the “green” chemistry trend, which is especially popular and promoted recently, and the developed method is the first one (according to the authors’ knowledge) based on halloysite in the determination of metal ions. The use of screen-printed sensors also allowed to reduce the volume of the analyzed solutions and eliminated the need for cleaning or special preparation of sensor surfaces. Stability studies indicated another advantage of the tested sensors: the possibility of their use for a relatively long time, as well as the use for individual sensors in multiple measurements. The preparation of the sensor is very simple and quick—it only requires mixing the individual components in the right proportions, applying them to a screen-printed electrode and drying in an air stream. All of the above show that MWCNTs/C_Hal/Nafion/SPCE sensors can be applied for analysis in a natural environment, which promotes the green analytical chemistry principles. The developed procedure of SPCE surface modification with the C-deposited halloysite and MWCNTs-based composite seems to be a good alternative to other sensors (see Table 3). The application of this sensor allowed for an accurate and precise determination of Cd^2+^ and Pb^2+^ ions in tap and well waters as well as in food samples, such as mineral water and honey, with good recovery. It is worth highlighting that the proposed electrode modification nanocomposite has great potential to be employed for the design and development of other (bio)sensors.

## Figures and Tables

**Figure 1 molecules-27-04608-f001:**
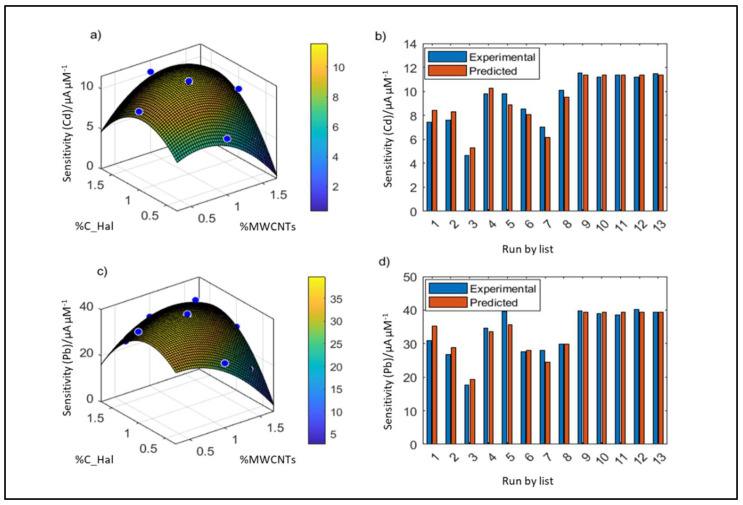
Response surface plots and model evaluation (predicted/experimental) for the optimization of the content of carbonized halloysite and multi-walled carbon nanotubes: (**a**) sensitivity of the Cd determination method; (**b**) comparison of the experimental and model-predicted sensitivity (Cd); (**c**) sensitivity of the Pb determination method; (**d**) comparison of the experimental and model-predicted sensitivity (Pb).

**Figure 2 molecules-27-04608-f002:**
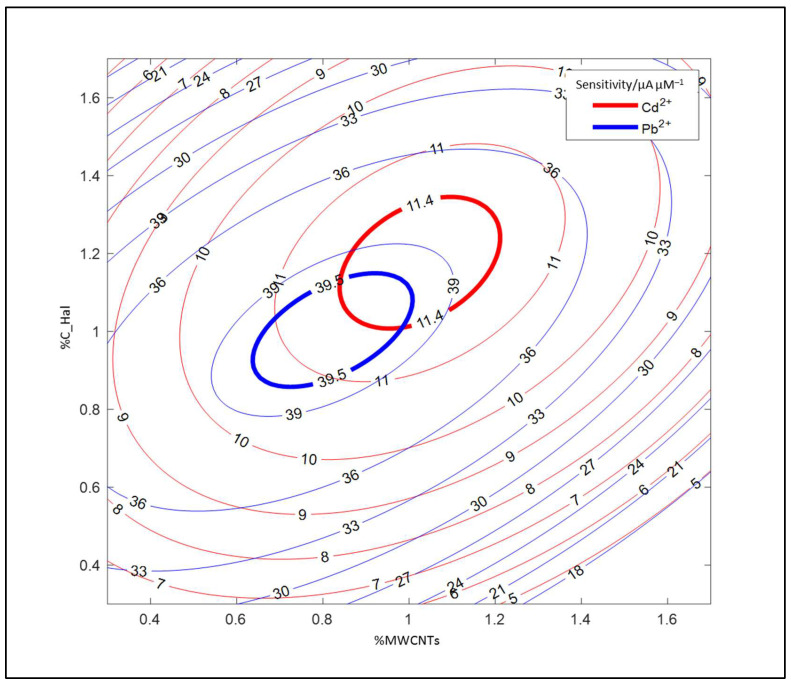
Projection of the model response surfaces for cadmium and lead determination.

**Figure 3 molecules-27-04608-f003:**
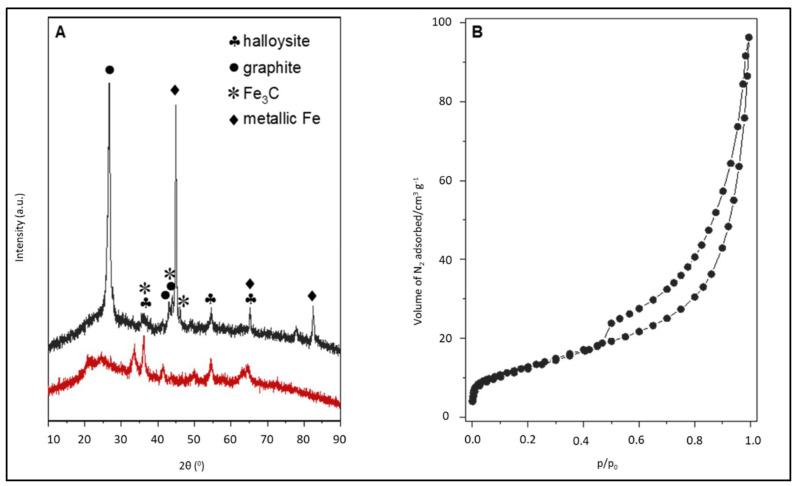
(**A**): X-ray diffraction patterns of natural halloysite (red line) and this clay after carbon deposition (black line), (**B**): N_2_ adsorption–desorption isotherms of carbon-deposited halloysite.

**Figure 4 molecules-27-04608-f004:**
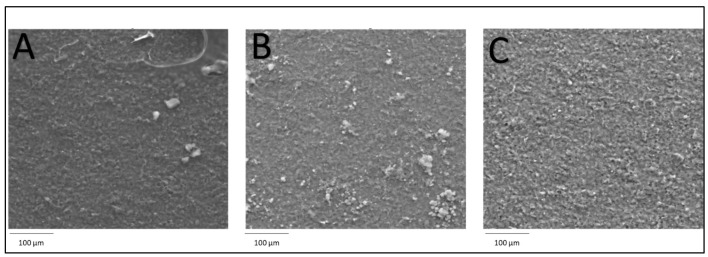
SEM images of unmodified electrode (**A**), C_Hal/Nafion/SPCE (**B**) and MWCNTs/C_Hal/Nafion/SPCE (**C**).

**Figure 5 molecules-27-04608-f005:**
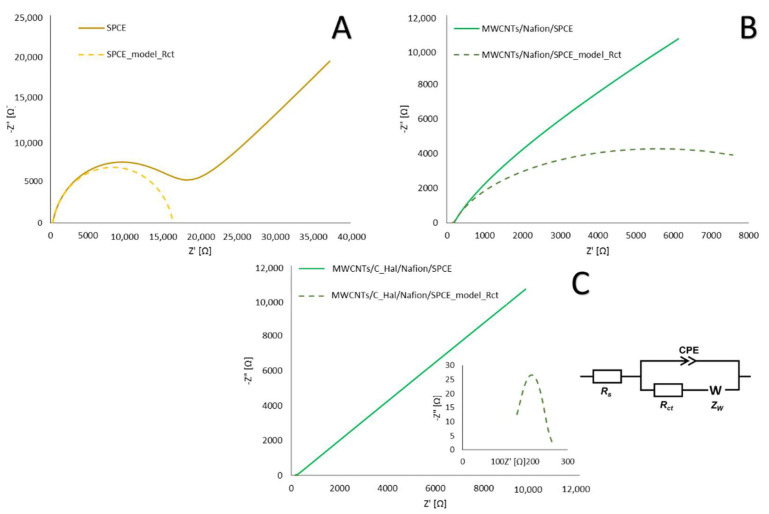
The impedance spectra recorded for [Fe(CN)_6_]^3−\4−^ (0.5 mmol L^−1^) in 1 mol L^−1^ KCl solution for (**A**) unmodified SPCE, (**B**) MWCNTs/Nafion/SPCE, (**C**) MWCNTs/C_Hal/Nafion/SPCE and model semicircle with the equivalent circuit diagram.

**Figure 6 molecules-27-04608-f006:**
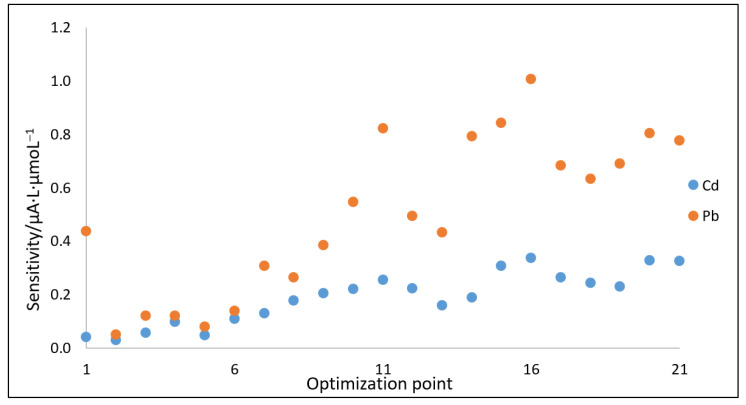
Change in sensitivities obtained from calibration dependencies during optimization process (for details see text above and Appendix A).

**Figure 7 molecules-27-04608-f007:**
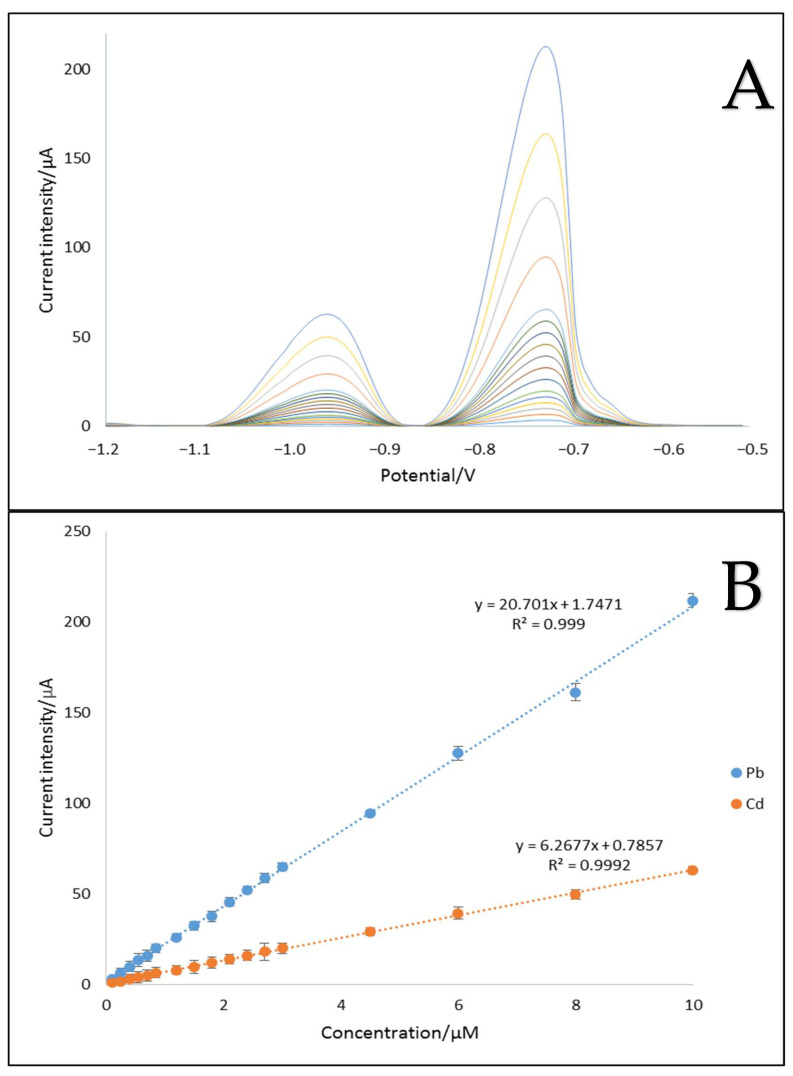
Differential pulse anodic stripping voltammograms (**A**) obtained upon increasing metal ions concentration in 1 mol L^−1^ acetate buffer solution (pH 4), following deposition at −1.2 V for 90 s and corresponding calibration curves (**B**) in the linear range from 0.1 to 10 μmol L^−1^.

**Table 1 molecules-27-04608-t001:** Sample space—the set of the experiments to be performed by a two-factorial central composite design.

Run	Actual Value of Factor % MWCNTs	Actual Value of Factor % C_Hal	Coded Value of Factor *x*_1_	Coded Value of Factor *x*_2_
1	0.5	0.5	−1	−1
2	0.5	1.5	−1	1
3	1.5	0.5	1	−1
4	1.5	1.5	1	1
5	0.3	1	−1.414	0
6	1.7	1	1.414	0
7	1	0.3	0	−1.414
8	1	1.7	0	1.414
9	1	1	0	0
10	1	1	0	0
11	1	1	0	0
12	1	1	0	0
13	1	1	0	0

**Table 2 molecules-27-04608-t002:** The comparison of electrochemical characteristics of tested electrodes based on parameters achieved from EIS measurements (*n* = 3).

Sensor	*C_dl_*/µF cm^−2^	*R_ct_*/kΩ	*σ*/kΩ s^−1/2^	*A_eff_*/mm^2^	*k_s_*/m s^−1^
SPCE	20.8 ± 0.1	16.3 ± 0.2	8.50 ± 0.07	3.30 ± 0.05	(9.9 ± 0.1) × 10^−6^
MWCNTs/Nafion/SPCE	8200 ± 50	10.80 ± 0.08	6.90 ± 0.04	4.07 ± 0.07	(1.21 ± 0.04) × 10^−5^
MWCNTs/C_Hal/Nafion/SPCE	9.68 ± 0.05	0.057 ± 0.002	3.87 ± 0.03	7.25 ± 0.04	(1.27 ± 0.01) × 10^−3^

**Table 3 molecules-27-04608-t003:** Comparison of analytical determination of Cd^2+^ and Pb^2+^ using reported electrochemical sensors.

Electrode	Technique	Linearity Range /µmol L^−1^	LOD /μmol L^−1^	Sample	Ref.
The fullerene C60-chitosan modified GCE	DPASV	Pb: 0.005–6.0 Cd: 0.5–9.0	Pb: 0.001 Cd: 0.021	milk, honey	[33]
Graphene/(AuNPs)/[Ru(NH_3_)_6_]^3+^/Nafion/(GCE)	ASV	Pb: 0.036–7.19 Cd: 1.5–11.27	Pb: 0.0012 Cd: 0.0011	meat, tuna, mushrooms, canned sardines	[34]
Bismuth-coated GCE	SWASV	Pb: 0.007–0.965 Cd: 0.013–1.779	Pb: 0.007 Cd: 0.356	rices	[35]
Polymer-coated BiFEs/GCE	SWASV	Pb: 0.010–0.290 Cd: 0.018–0.534	Pb: 0.010 Cd: 0.018	rock, urine	[36]
Diacetyldioxime -CPE	AdSV	Pb: 0.10–1.0 Cd: 0.25–2.5	Pb: 0.01 Cd: 0.04	water	[37]
ZnFe_2_O_4_/GCE	DPASV	Pb: 0.048–0.627 Cd: 0.089–1.156	Pb: 0.005 Cd: 0.022	wastewater	[38]
MC/Nafion/GCE	SWASV	Pb: 0.024–0.338 Cd: 0.044–0.623	Pb: 0.0006 Cd: 0.0004	soil	[39]
BiONPs-CS-GCE	DPASV	Pb: 0.4–2.8 Cd: 0.8–5.6	Pb: 0.15 Cd: 0.05	tap water	[40]
NanoSiO_2_/BiFEs/GCE	SWASV	Pb: 0.010–0.724 Cd: 0.018–1.334	Pb: 0.001 Cd: 0.005	water	[41]
MWCNTs/C_Hal/Nafion/SPCE	DPASV	Pb/Cd: 0.10–10	Pb: 0.0051 Cd: 0.0106	tap water, mineral waters, well water, honey	This work

GCE—glassy carbon electrode, AuNPs—gold nanoparticles, BiFEs—bismuth film electrode, CPE—carbon paste electrode, MC—mesoporous carbon, BiONPs—bismuth oxide nanoparticles, CS—chitosan.

**Table 4 molecules-27-04608-t004:** Results of Cd^2+^ and Pb^2+^ determination in spiked samples.

Sample	Amount Added/nmol L^−1^	Amount Obtained ^a^/nmol L^−1^	Recovery/%	RSD/%
Tap water	250	248	99.2 ± 0.6	3.5
	750	751	100.1 ± 0.8	4.2
Well water	250	244	97.6 ± 0.5	2.8
	750	742	98.9 ± 0.7	3.8
Mineral water 1	250	247	98.8 ± 1.3	6.9
	750	742	98.9 ± 0.8	4.5
Mineral water 2	250	254	101.6 ± 0.7	3.9
	750	760	101.3 ± 0.5	2.7
Honey	250	241	96.4 ± 1.6	8.5
	750	739	98.5 ± 1.2	6.6

^a^ mean of three measurements.

## Data Availability

Not applicable.

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
