# Peer review of "Electrochemical Sensing of Pb2+ and Cd2+ Ions with the Use of Electrode Modified with Carbon-Covered Halloysite and Carbon Nanotubes"

_molecules, 2022, doi:10.3390/molecules27144608_

Round 1
Reviewer 1 Report
The manuscript reports preparation of modified electrode using carbon-covered halloysite and carbon nanotube and its application for electrochemical sensing. The following concerns should be addressed.
1. The plots of intensity of oxidation currents versus the root of scan rate should be given.
2. The process of calculating the electrochemical active surface areas should be elaborated in more details.
3. What does CMK-3 stand for?
4. SEM images are not very clear. Better images should be given and placed in the main text.
5. Have reproducible tests been performed for the DPASV experiments? Error bars should be added in Figure 6.
6. EDX spectra is mentioned in the method part. But no EDX images are given throughout the manuscript.
Author Response
Thank you very much for all your comments, which are very valuable and help to improve the publication. All changes to the main text are marked in yellow.
Please see the attachment with the answers of Your comments.

Reviewer 2 Report
The manuscript “Electrochemical Sensitivity of Pb2+ and Cd2+ Ions with the Use of Electrode Modified with Carbon-Covered Halloysite and Carbon Nanotubes” presents optimization of sensor for lead and cadmium cations detection. The sensor was prepared using screen printed carbon electrode covered with carbon-modified halloysite and carbon nanotubes binded with Nafion. The optimized sensor presented a wide response range and LOD sufficient for practical application.
The quality and novelty of presented research is sufficient for publication in the Molecules journal, however there are some minor issues with the manuscript.
1. Overall, there are too many grammatical errors, and the text is written in a convoluted manner, making it hard to follow in some sections, e.g., line 52, “On the one hand” should be used if “on the other hand” is used in line 54, “equals” line 109, “it was decided to choose a mixture” line 110, “which is a typical activity in the context of tasks aimed at developing new sensors”, lines 159-162, etc.
I would suggest avoiding long and overly complex sentences, as most of the readers of the journal are not native English speakers.
2. I would suggest placing the experimental section before the results in the manuscript. The experiment was complex and jumping between the main text, the experimental section and appendix makes reading the manuscript a very tedious task.
3. All figures presented in the manuscript should be unified, both graphically (font type, use of a frame), as well as in terms of how the axis title and unit is used (using “/” or “[]”). Also, in the Figure 5 the points should not be connected in my opinion. If possible, the Figures and Tables should not be divided between two pages.
4. In my opinion selected CV measurements in the presence of lead and cadmium cations should be presented in the manuscript or in the appendix.
5. Not all abbreviations are explained in Table 3.
6. It is not clear how the EIS fitting was done, and why it was limited only to high frequency of the spectrum. Was the same model used for all fittings? Please provide a discussion how each element of the model corresponds to the actual sensor.
7. The scale in Fig. A2 is too small. Why some data is missing in the Table A2?
Author Response
Thank you very much for all your comments, which are very valuable and help to improve the publication. All changes to the main text are marked in yellow.
Please see the attachment with the answers to Your comments.

Reviewer 3 Report
Review Report
Manuscript ID: molecules-1779449
Title: Electrochemical Sensing of Pb2+ and Cd2+ Ions With The Use of Electrode Modified with Carbon-Covered Halloysite and Carbon Nanotubes
Journal: Molecules
In this extensive study, an advanced characteristic of a new electrochemical (voltammetric) method for the sensitive and selective determination of Cd and Pb ions has been presented. The use of screen-printed carbon electrodes modified with carbon-deposited natural halloysite (C_Hal) and multi-walled carbon nanotubes (MWCNTs) was developed. The proposed electrochemical sensor was characterized by using electrochemical impedance spectroscopy and cyclic voltammetry (CV). The morphology and structural characteristics were investigated by scanning electron microscopy and X-ray powder diffraction. The selection of the most desirable and efficient composition of nanocomposite was done by a two-factorial central composite design. The Nelder-Mead simplex method was chosen for obtaining the optimal measuring parameters of differential pulse anodic stripping voltammetry used for quantitative analysis. The proposed sensor was effectively applied for the determination of metal ions in different natural water and honey samples with a high recovery of 96.4–101.6%.
Reviewer’s suggestions:
Line 42 Correct the proper marking of the reference number.
Line 113 to 127 Author had e to give some more detailed explanation about how did they calculate the electrochemical active surface areas by using the Randles-Ševčik equation.
Line 115 Correct power -6.
Line 121/to 127 and 423 Authors have to give some explanation about the difference between the geometric area of the working electrode 7.07 mm2 and the electrochemically obtained surface area. Why the electrochemically obtained surface area in two cases is smaller than the geometric. It would be logical that the electrochemically surface area is larger than the geometric area. So please give some reasonable explanation.
Line 242 Table 2. The increase in Cdl from 20.8 mF cm-2 for SPEC to 8200 mF cm-2 for MWCNTs/Nafion/SPCE is logical but then Cdl decreased to 9.68 mF cm-2 for MWCNTs/C_Hal/Nafion/SPCE, and this is something that is not logical. A smooth electrode surface has Cdl of about 20 mF cm-2. An increase in real surface area will also increase Cdl. Authors have to check the value of Cdl, I suspect that it has to be larger than Cdl for MWCNTs/Nafion/SPCE. If not, the authors have to provide some reasonable explanation. Also, the surface roughness may be estimated as R =Cdl/20 mF cm-2. It characterizes the ratio of the real (true, electrochemically area) to the geometrical surface area. This is something that authors have to consider (Ref: Karimi Shervedani R, Lasia A. Kinetics of hydrogen evolution reaction on nickel-zinc–phosphorous electrodes. J Electrochem Soc 1997;144:2652–7). It will be also of great interest to provide the equation form calculation of Cdl like in Electrocatalytic activation of Ni electrode for hydrogen production by electrodeposition of Co and V species January 2009 International Journal of Hydrogen Energy 34(2):703-709, DOI: 10.1016/j.ijhydene.2008.09.024, Milica Marceta KaninskiMilica Marceta KaninskiVladimir M. NikolicVladimir M. NikolicGvozden S. TasicGvozden S. TasicZlatko Lj. Rakocevic.
This study has the potential to be cited.
I recommend to the Editorial office to consider this manuscript for publication, but after major revision mostly due to the relatively poor interpretation of the electrochemical characteristics such as the Cdl and the real and geometric surface area.
Author Response
Thank you very much for all your comments, which are very valuable and help to improve the publication. All changes to the text are marked in yellow.
Please see the attachment with the answers to Your comments.

Round 2
Reviewer 3 Report
The authors addressed all my comments. I recommend this manuscript for publication.